# A Framework to Simulate Friction Stir Additive Manufacturing (FSAM) Using the Finite Element Method

**DOI:** 10.3390/mi15030303

**Published:** 2024-02-23

**Authors:** Bahman Meyghani, Reza Teimouri

**Affiliations:** 1BKL B.V., Collse Heide 1, 5674 VM Nuenen, The Netherlands; bahman.malaki.2009@gmail.com; 2Department of Mechanical Engineering, Faculty of Engineering Technology and Built Environment, UCSI University, Taman Connaught, Kuala Lumpur 56000, Malaysia; 3Chair of Production Engineering, Faculty of Mechanical Engineering, Cracow University of Technology, John Pawła II, 31-864 Cracow, Poland

**Keywords:** friction stir additive manufacturing (FSAM), solid-state, manufacturing technology, thermomechanical behavior, frictional behavior, Eulerian

## Abstract

Defining an accurate friction model without having the mesh distortion in an optimized computational time has always been a significant challenge for modelling solid-state natural processes. The presented paper proposes an Eulerian frictional-based solid static model for the accurate modeling of sliding and sticking conditions for the friction stir additive manufacturing process (FSAM). For the frictional behavior, a modified friction model is proposed to investigate the sliding and sticking conditions during the process. The magnesium alloy is selected as the workpiece material and AZ31B-F is employed as the filler material. Two different subroutines, Dflux and Sfilm, are used in order to simulate the heat flux during the process. The convection and emission during the process are determined using the Goldak double ellipsoidal model. DC3D8 and C3D8R elements are employed as the thermal and mechanical models, respectively. The results indicated that the temperature sharply increased up to 870 °C in the first and the second layers. After that, the increasing rate becomes slower with a maxim temperature of 1310 °C. A linear cooling behavior is obtained at the cooling step. The stress results indicated that the tool and the filler material pressure play a significant role in increasing the stress at the center of the workpiece. On the sides of the workpiece, a peak stress is also obtained due to the clamping force. At the cooling phase for the center of the workpiece, the longitudinal residual stress of 5 MP and transverse residual stress of 7 MPa (compression) are achieved. The distortion of the workpiece is also investigated and a maximum value of 0.13 mm is obtained. To wrap up, it should be noted that by implementing an accurate sliding/sticking condition in a frictional based model, a more comprehensive investigation about frictional interactions and their influence on thermal and mechanical behavior can be carried out.

## 1. Introduction

Enhancing the quality of manufactured products requires the development of advanced manufacturing processes [1,2]. Magnesium plays a significant role in manufacturing technology due to its light weight and its excellent resistance ratio, etc.; however, it has a complicated structure, and therefore experimental investigation of it demands a lot of cost and time. Thus, a lot of researchers focused on the finite element modeling of this advanced manufacturing processes for magnesium alloys [3,4]. Friction stir additive manufacturing (FSAM) is one the advanced manufacturing processes that have been recently developed by many researchers [5,6]. The process requires a complicated thermomechanical and friction behavior. Accordingly, the experimental and numerical investigation of the process is still ongoing by many scholars [5,6].

Finite element modelling (FEM) is used to develop the solid-state nature of friction-based process modeling (the solid-state additive manufacturing process and solid-state friction stir welding) in a Lagrangian and Arbitrary Eulerian–Lagrangian (ALE) framework to investigate both elastic and plastic strains [7,8]. Aluminum and magnesium are used in these studies as the main materials. Moreover, the temperature is studied to validate the model. The results of the experimental and numerical simulation are compared. The results also indicated that the model has an acceptable accuracy. Denlinger et al. [9] studied the Effect of the stress relaxation on the distortion at the heating zone. The model theoretically yielded a good alignment for the microstructure and mechanical properties between the model and the literature [10]. A moving heat source [11] examined the heat generation and temperature evaluation during the additive friction stir deposition by Garcia et al. for aluminum and magnesium alloys. The Monte Carlo method is employed to study the recrystallization, the growth of the grains and the precipitate evolution validated by experimental tests (grain sizes and the hardness). It was found that the grain size at different layers of the process was highly affected by the temperature in the starting zone. To elaborate further, smaller average grain sizes raised the hardness at the top layers. Finite element method was used [12] to develop a layer-by-layer study for similar and dissimilar materials. In this research, a non-beam-based metal additive with manufacturing mechanical properties such as plastic deformation and temperature-dependent material properties is employed. For the finite element section, the Smoothed Particle Hydrodynamics (SPH) method is used to solve the mesh distortion during the simulation. Vickers hardness distribution is also used to validate the work experimentally. A good agreement between the presented work and experiments in terms of the stress distribution and the temperature was found. It is also claimed that friction model plays a significant role in the accuracy of the simulated model. The optimization of the FSAM process parameters was performed. The results indicated that the temperature-dependent thermal and mechanical properties affect the final temperature results. It was also shown that the friction model accuracy influences the results. It can be summarized that most of the literature mentioned above did not consider the friction model as a critical parameter in their studies.

Other literature [13,14,15] noted that implementing an accurate friction model affects the accuracy of the results for friction-based solid state processes, because friction and plastic deformation are the two key sources of heat during the process [16,17,18,19]. Furthermore, defining an accurate and precise friction model for solid-state processes has always been considered a challenge. On one hand, the friction model should be accurate and be able to simulate the frictional behavior especially at the points of interaction between the parts, and on the other hand, the simplicity of the friction model highly decreases the computational costs. Due to the importance of the frictional behavior, researchers have focused on investigating the frictional and plastic deformational behavior during frictional based solid-state natural processes. Frictional behavior is studied to optimize the process and to investigate the thermomechanical behavior. In this regard, some of the literature [20] employed the Coulomb friction model as their friction model for modelling aluminum and magnesium, as the Coulomb friction model can be accurate at lower temperatures [21,22,23,24]. To dig down deeper into the problem, the Coulomb model can be accurate when the sticking condition is available [25,26,27,28]. As long as the temperature rises, the sticking condition transforms into a sliding condition and this phase transformation leads into an overestimation of friction conditions for the Coulomb friction model [29,30]. Heat transfer is experimentally measured by researchers [25,26,27,28] to study the temperature behavior during these processes for the aluminum and magnesium alloys. The results are validated by comparing the experimental results with numerical findings. Material properties are examined by sensitivity analysis to determine the optimized parameters. Structural dynamic analysis is also carried out to validate and verify the results. An optimized model [3] is developed to reduce the error between the two different heat generation methods. Different frictional methods are implemented in the model. Young’s modulus and Poisson’s ratio are considered as two temperature-dependent processes. It is found that the total average error in terms of the heat generation is decreased. Meyghani and Awang [17] simulated a friction stir process in a fully thermomechanical model. The model was explicitly dynamic. Elastic and plastic strains are studied and the experimental tests are compared with the results to verify the model. Finally, a good agreement between the experiments and the simulation is found. The microstructure and mechanical properties of the WE43 alloy produced during the additive friction stir process is studied by Calvert et al. [18] in an integrated theoretically developed model. The microstructure and mechanical property results are compared to the present literature, and finally, a good agreement is obtained for the model. The heat source is modeled as a moving one to investigate the FSAM thermal behavior [19]. This study used a theoretical method to analyze the recrystallization, grain growth, precipitate size distributions and mechanical properties. In order to validate the results, experimental tests were organized in terms of the grain size and the hardness. It was found that the grain size at the layers is highly dependent on the stirring zone temperature. Thus, the quality of the work required a highly accurate prediction of the temperature. Some of the literature optimized the process [20,21] to study the shear deformation during the friction stir welding of aluminum and magnesium alloys. The classical Coulomb frictional model was used to simulate the frictional behavior. High mechanical properties for the process were achieved. Compared with all of the parameters, conduct material selection was found to be the best way to investigate the highest material properties [23]. Innovative manufacturing processes [15] for metallic components were conducted for studying the disruptive and innovative manufacturing processes. It can be claim that the best mechanical properties for the results are found in comparison with the present literature [23]. A physical-based simulation was performed to study the defect formation and material flow during the friction-based solid-state processes [24]. The results were validated and verified by the experimental data. This study investigated the material disposition layer-by-layer for similar and dissimilar materials. Plastic deformation and temperature were employed to confirm the model.

Another significant challenge for modelling friction-based solid-state natural processes is the mesh distortion during the process [17,31,32]. Handling the mesh distortion using the Lagrangian mesh model was challenging because of the large plastic deformation happening during the simulation [33]. Most of the mesh sensitivity analysis showed that arbitrary Lagrangian–Eulerian (ALE) highly increased the computational costs [13,34]. In this regard, the Eulerian model was found to be the optimal model for handling the mesh distortion with an appropriate simulation cost [35]. In this method, the mesh remains fixed, and the material moves freely between the elements. Smoothed particle hydrodynamics, called the SPH model, was used to simulate the temperature distribution. However, the computational costs of this method were higher than the previously mentioned methods [13]. 

Reviewing the literature showed that the definition of the friction model was one of the key parameters for predicting the heat generation during the process. To simplify the friction behavior, most of the literature used the Coulomb friction model for the FSW and FSAM processes. In this regard, it should be noted that sticking and sliding conditions appear for the process. Therefore, there is a need to develop a model to successfully simulate the sticking/sliding behavior. Moreover, the definition of an optimized mesh model should be considered as a critical task in order to solve the mesh distortion problem and save on computational costs and time. 

This paper presents a novel method by implementing a modified friction model to address the challenges of the sliding and sticking conditions encountered during the modelling of friction stir additive manufacturing (FSAM) processes. The developed friction model introduces a significant development in the modelling field, offering improved accuracy and reliability in depicting the complex interactions between tool, the filler material and the workpiece materials. This modified friction model is seamlessly integrated into an Eulerian-based computational solid mechanical model (CSM), that serves as the computational framework for studying the thermomechanical behavior of FSAM. By leveraging the abilities of the CSM, the model enables a comprehensive analysis of the thermal and mechanical responses of the workpiece under different process conditions. It should be mentioned that, in addition to the frictional interactions, the model is able to accurately simulate the convection and emission. In this regard, convection and emission are calculated using the renowned Goldak double ellipsoidal model. These thermal calculations are integrated into a 3D finite element model through the utilization of specialized subroutines, namely Dflux and Sfilm, ensuring a holistic representation of the thermal environment within the workpiece during FSAM. The application of this integrated method results in a comprehensive investigation of the thermomechanical behavior of FSAM. The model also is able to analyze the temperature distributions and stress fields. Furthermore, the model facilitates the prediction and characterization of distortions in the final workpiece, aiding in the optimization of process parameters and the enhancement of product quality in FSAM applications. 

## 2. Materials and Methods

### 2.1. Heat Governing Equations

The entire heat that is generated during the FSAM process can be explained as below [13,36,37],
(1)$$Q_{Total}=Q_{Friction}+Q_{Plastic}$$

Basically, this entire heat is created by the different sections of the tool including the shoulder tip surface (*ST*), filler material tip surface (*FT*) and filler material side surface (*FS*) [38],
(2)$$Q_{Total}=Q_{ST}+Q_{FT}+Q_{FS}$$

So, this heat can be estimated as [39],
(3)dQ=ω.dM [J/s]
where ω is the tool rotational velocity (rad/s) and *M* is the torque [40]. 

Torque can be considered as the force (*F*) multiplied by the radius distance (*r*) and the force can be considered as the cross product of shear stress (τ) and area (*A*). Therefore, Equation (3) can be written as [41],
(4)dQ=ω.r.τ.dA

Finally, the entire heat equation can be written as below,
(5)$$\frac{\partial }{\partial x}\left(k_{x}\frac{\partial T}{\partial x}\right)+\frac{\partial }{\partial x}\left(k_{y}\frac{\partial T}{\partial y}\right)+\frac{\partial }{\partial x}\left(k_{z}\frac{\partial T}{\partial z}\right)+q=\rho c\frac{\partial T}{\partial t}$$

The Goldak double ellipsoidal model is used to calculate the heat. The front semi-ellipsoids (Equations (6) and (7)) are employed to calculate the heat flux at the back and the front side.
(6)$$Q_{r}\left(x,y,z\right)=\frac{6\sqrt{3}f_{r}Q_{Total}}{a\;b\;c_{r}\;\pi \sqrt{\pi }}e^{\left(-3x^{2}/a^{2}\right)}e^{\left(-3y^{2}/b^{2}\right)}e^{\left(-3z^{2}/c_{r}^{2}\right)}$$
(7)$$Q_{r}\left(x,y,z\right)=\frac{6\sqrt{3}f_{f}Q_{Total}}{a\;b\;c_{f}\;\pi \sqrt{\pi }}e^{\left(-3x^{2}/a^{2}\right)}e^{\left(-3y^{2}/b^{2}\right)}e^{\left(-3z^{2}/c_{f}^{2}\right)}$$
where, $c_{r}$, $c_{f}$, *b*, and a are constants and $f_{r}$ and $f_{f}$ can be estimated as,
(8)$$f_{r}+f_{f}=2$$

Tool transverse velocity is considered as *v* in the *z* direction, hence,
(9)$$z^{\prime }=z-v.t$$

After substitution of $z^{\prime }$ as a replacement for of *z* in the equation, the heat will be aligned at the *z*-direction with respect to the time and the velocity. 

Moreover, the heat transfer throughout the process is considered as a transient heat transfer phenomenon. Therefore, the time-dependent parameter’s nature is considered for the process. It is significant to mention that the heat at the workpiece plays a crucial role in the quality of the process and the heat transfers to the tool plays a critical role in the tool life. 

### 2.2. Finite Element Model Description

The Dflux subroutine is used to implement the Goldak heat input model and the Sfilm subroutine is employed to model the convection and radiation during the heat transfer. The initial parameter is defined for the beginning of the process. After that, the outcome of the previous layer is used as an input for the next layer. The workpiece sizes are 153 mm in width, 203 mm in length and 6.35 mm in thickness. AZ31B-F is used as the filler material. Additionally, a clamping force is considered as a boundary condition for the model. For the thermal model DC3D8 (An 8-node linear heat transfer brick) and for the mechanical model C3D8R (An 8-node linear brick, reduced integration and hourglass control) are used. Temperature-dependent mechanical and thermal properties always provide higher accuracies during the solid-state processes. As a result, temperature-dependent material properties that are implemented as an initial input are explained in Table 1, Table 2 and Table 3.

Based on the literature [36,37], a transverse speed of 203 mm/min, rotational speed of 400 RPM, axial force of 7750 N, shear stress of 126 MPa and a tool diameter of 27.8 mm are employed as the simulation input. 

### 2.3. Friction Model Governing Equations

A modified friction model (Equation (10)) is used in this study [27] as below,
(10)$$\tau =-P\cdot g\left({∆v}_{g}\right)\cdot \frac{{∆v}_{g}}{\left|{∆v}_{g}\right|}$$

In order to split up the heat by friction and plastic deformation, Equations (11) and (12) are introduced as below,
(11)$$\dot{q_{f}}=\left({∆v}_{g}\right)\cdot \tau$$
(12)$$\dot{q_{pl}}=\eta \sigma :{\dot{\varepsilon }}_{pl}$$

Furthermore, the heat during the process is dependent on the two types of heat sources. Frictional heat caused by the relative velocity (${∆v}_{g}$) and shear stress (τ), and the plastic deformation heat caused by the material deformation that depends on the stress (σ) and true strain (ε). Equation (10), that is the modified friction model, incorporates the relative velocity (${∆v}_{g}$) and pressure (P) into the calculation, which is more accurate than the Coulomb model that only depends on the constant frictional coefficient and the pressure.

## 3. Results and Discussions

### 3.1. Temperature History

The temperature history results at different layers for the model is described in Figure 1. As can be seen, at the initial step, a sharp increase up to 870 °C in the temperature is observed (at the first and the second layers). This increase is caused by the high pressure from the tool axial force and the filler material [38,39]. Due to the combination of the rotational movement and the axial pressure, deformation of the material happens, and this energy converts to heat, hence an increase in the temperature occurs. Another reason for the heat is the localized heating that is caused by friction. After moving the material along the tool path, the material deposition process occurs layer-by-layer, and this deposition creates a localized heating [38,39]. As can be seen, a maximum value of 1310 °C is obtained at the last layer. Furthermore, during the cooling phase, an almost linear behavior is obtained. 

### 3.2. Temperature at the Layers

As can be seen in Figure 2, an increase in the temperature up to 870 °C is achieved during the first layer. This temperature transfers the material phase from solid to the softened solid state. At the beginning of the process near the tool, a sticking condition between the workpiece and the tool is generated. This issue caused the conversion of mechanical energy into heat. Hence, a very sharp and fast increase in the temperature up to 870 °C occurs. As the process continues, the temperature decreases because of the prior material preheating at the previous layer, therefore a uniform temperature distribution is achieved. As a result, an onion ring structure is shaped at the workpiece. As can be seen, as long as the distance from the center line increases, the size of the rings become larger. The results of the temperature at the first layer are also confirmed by the literature [39,40,41]. It should be noted that this frictional heat is significant for softening the material and assisting the plastic deformation. It can be also seen from Figure 2 that, due to the implementation of the Eulerian modelling method, there is no mesh distortion, and the job could continue smoothly with an optimized computational cost [39,40,41]. 

The temperature at the second layer of the process has an almost similar behavior to the first one (see Figure 3). The only difference is the preheating caused by the first layer, which results in a smoother surface and the temperature behavior at the second layer. In addition, this issue caused a wider range for the temperature across the workpiece. It should be noted that, near the tool, the sticking friction behavior is obtained while far from the tool the sliding behavior appears for the model; therefore, the friction model should be able to optimize the sliding and sticking conditions at a same time. Moreover, the pressure from the filler material assists in increasing the temperature up to 1100 °C. This phenomenon shows up as a thermal interplay between the continuous layers during the additive manufacturing process [41,42,43]. 

Figure 4 indicates the temperature distribution at the workpiece in the third layer of the process. The maximum temperature at this layer increases up to 1250 °C, which is a little bit higher than the previous step. Hence, following the layers, a temperature decrease is observed at the workpiece; however, as long as the layers are added to each other, the increasing rate declines. As can be seen, the main concentration of the peak temperature is the center of the layer, beneath the filler material. Like the previous layer, a sticking condition occurs during the interaction between the workpiece and the tool. It is notable that during the sticking condition some parts of the materials adhere to the tool. Providing the layers are added to each other, a larger heat zone at the workpiece is observed because of the preheating at the initial steps [42,43,44,45]. As with the previous layer, a sliding condition is also present far from the center of the workpiece. Furthermore, the interaction between the layers and the surrounding material require a sliding behavior. It can be seen from the figure that as the distance from the center of the workpiece increases, the size of the rings increases as well. Thus, the sliding/sticking behavior is indicated in the simulation. 

In Figure 5, the temperature profile at the last layer (fourth layer) of the workpiece is shown. From the previous layer, a temperature of 900 °C remains; hence, when compared to the previous layers, a larger heat-affected zone is observed. As the process continues, a decrease in the temperature, due to the heat concentration primarily under the tool, is obtained (due to the sticking behavior). This temperature remains up to 1000 °C (below the material’s melting point). It should be said that the deposition of the material at this phase depends on the heat generated by the friction. This confirms the importance of the highly accurate friction model. It is important to note that the axial pressure on the tool decreases due to the softening of the material that reduces the interlayer adhesion; thus, the integrity of the additively manufactured part is observed. It can be mentioned from the model that, due to the implementation of the Eulerian mesh, the job could smoothly continue up to the last step of the simulation. This indicates that the model’s computational costs are optimized. The literature [31,32] reported that ALE mesh can simulate the process; however, the computational costs and the mesh distortion are critical challenges that require a lot of computational costs, especially at the first layers of the process. In contrast, using the Eulerian model decreases the computational costs, and shows a similar pattern for the simulation. As can be seen, in order to optimize the model, the mesh sizes at the center of the workpiece are larger than at the sides [42,43,44,45]. 

### 3.3. Cooling Phase Results

The cooling phase is indicated in Figure 6. As can be seen, there is a smooth cooling behavior at the surface of the workpiece. Since the material at the center of the workpiece faces a disparity, its cooling rate is faster than the other parts. The most important reason for this disparity is the higher material velocity at the center of the workpiece that results in lower heat generation and a higher cooling rate [43,44,45,46,47]. In the cooling phase, this thermal behavior appears across the workpiece, and this causes larger rings (see Figure 6). The onion rings that appear during the cooling phase provide a significant investigation into the thermal history that indicate and leverage observations and assist in optimizing the process performance and reliability. 

### 3.4. Stress Distribution Results

The results of the stress are shown in Figure 7. It is obvious that the parts that face higher temperatures have the highest stress, and this issue can be completely observed at the center of the workpiece. The most significant reason for this issue is the pressure from the tool, and especially the shoulder on the workpiece. Additionally, the filler material pressure on the workpiece causes this peak stress at the center of the workpiece. On the sides of the workpiece there are also some stress results that occur because of the clamping boundary condition. Furthermore, a boundary condition is defined as the clamping of the workpiece. As the process continues, the workpiece tends to have a concave shape, and the clamping force resists against this trend; thus, a high stress is observed at the sides of the workpiece [47,48,49,50]. Figure 8 indicates the longitudinal stress distribution during the process. As can be seen, the values at the center of the workpiece increases up to 5 MPa during the cooling stage. Moreover, a “W” shape behavior is obtained for this stress. This remaining stress could influence the properties and the material behavior. Figure 9 shows the transverse residual stress at the workpiece cross section during the cooling step. The values are around 7 (compression) MPa at the center of the plate. As can be seen, an “M” shape behavior is obtained across the workpiece cross section. This shows that at the sides of the workpiece higher stress is obtained in comparison to the center [47,48,49,50]. 

### 3.5. Distortion Results

The results of the distortion are indicated in Figure 10. As can be seen, the center of the workpiece is deformed around 0.13 mm. Moreover, a convex shape is formed due to the spring back compensation caused by the pressure from the tool and the filler material. Modelling this distortion is challenging, because in the model, a heat flux is considered as the tool, and thus, an input pressure should be considered as the initial tool pressure. Following this, the results of the previous step should be used as an input for the next layer. Hence, the accuracy of the friction model is notable. It can be seen that the distortion at the sides of the workpiece is less than at the center of the workpiece. This occurs because of the clamping. Furthermore, a boundary condition is fixed the workpiece sides; hence, minimum distortion is observed at the center of the workpiece [51,52]. 

## 4. Conclusions

In this study, a complicated friction model is employed in an Eulerian framework to study the thermomechanical behavior of the friction stir additive manufacturing process (FSAM). The results showed a significant influence of the sliding/sticking conditions on the accuracy of the model. The maximum temperature at the beginning of the process sharply reaches up to 870 °C, and after that, the increasing rate becomes smoother with the maximum peak temperature of 1310 °C. It is summarized that the preheating from the previous layers affects the subsequent peak temperature. Moreover, a faster linear cooling rate is observed at the center of the workpiece. During the cooling step, a maximum value of 5 MP and 7 MPa (compression) for the longitudinal and transverse residual stress is obtained, respectively. It is also concluded that, due to the use of Eulerian mesh, the mesh distortion problem is solved with an optimized computational cost. Furthermore, a higher distortion at the center of the workpiece is observed. To summarize, the proposed Eulerian frictional-based solid static model resulted in an accurate representation of the sliding and sticking conditions, improvement of predictive capabilities, enhancement of the process control, potential design optimization and a contribution to the fundamental understanding of the process in more detail.

## Figures and Tables

**Figure 1 micromachines-15-00303-f001:**
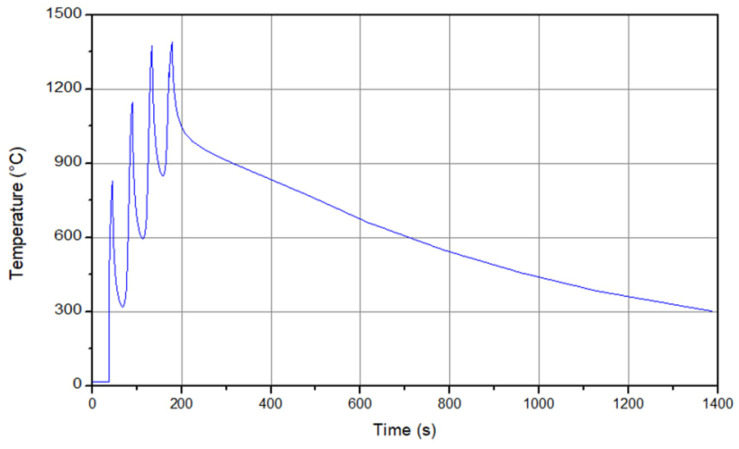
Temperature history for different layers during the process.

**Figure 2 micromachines-15-00303-f002:**
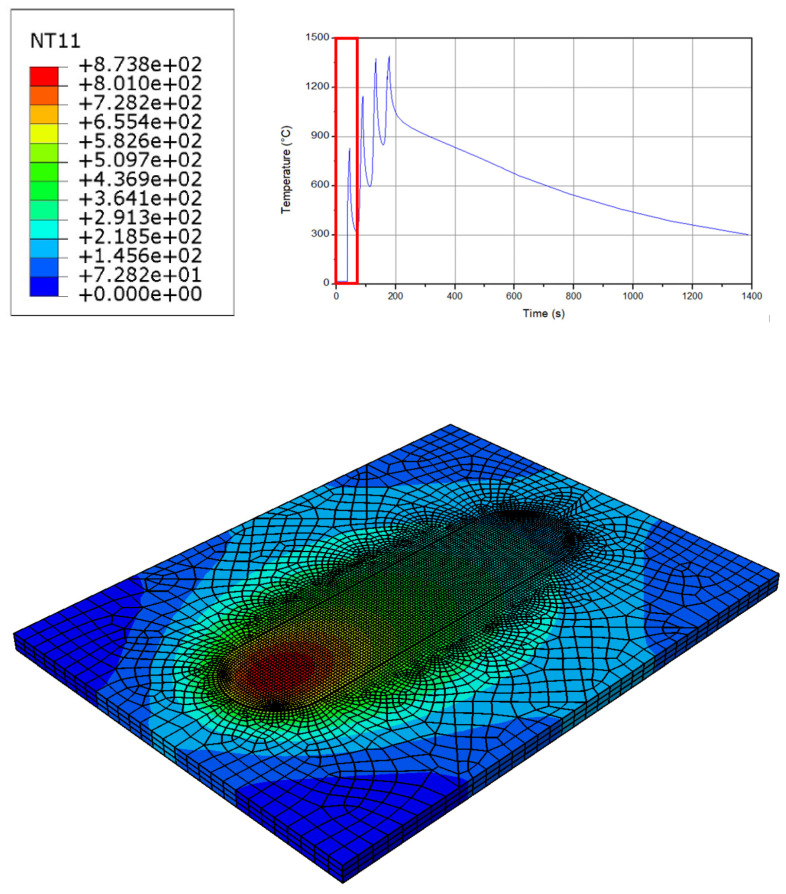
The temperature distribution at the workpiece in the first layer.

**Figure 3 micromachines-15-00303-f003:**
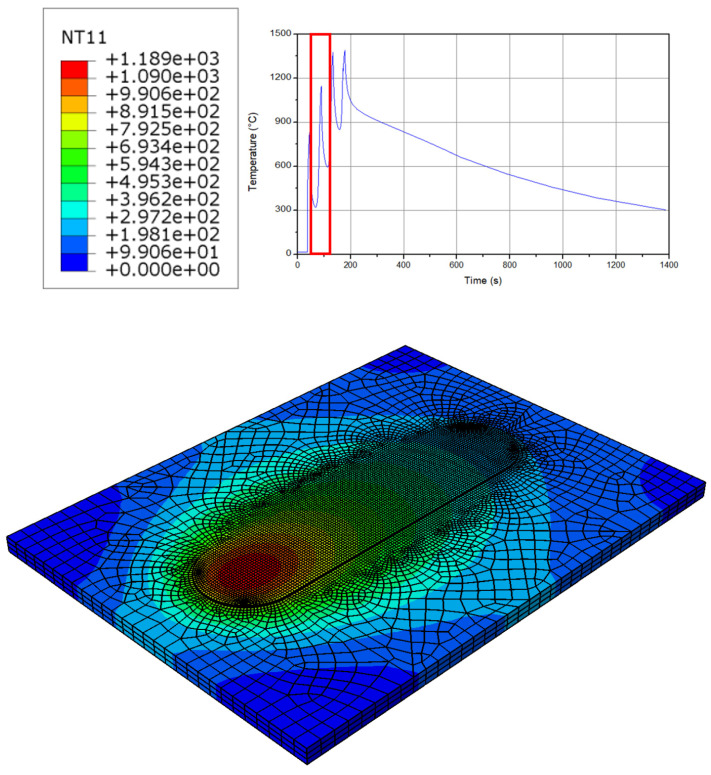
The temperature distribution at the workpiece in the second layer.

**Figure 4 micromachines-15-00303-f004:**
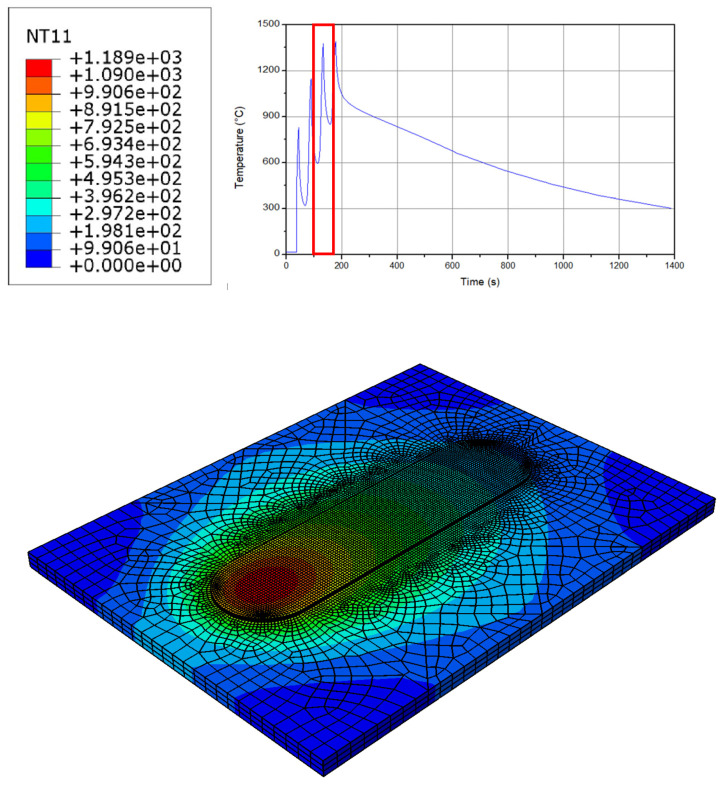
The temperature distribution at the workpiece in the third layer.

**Figure 5 micromachines-15-00303-f005:**
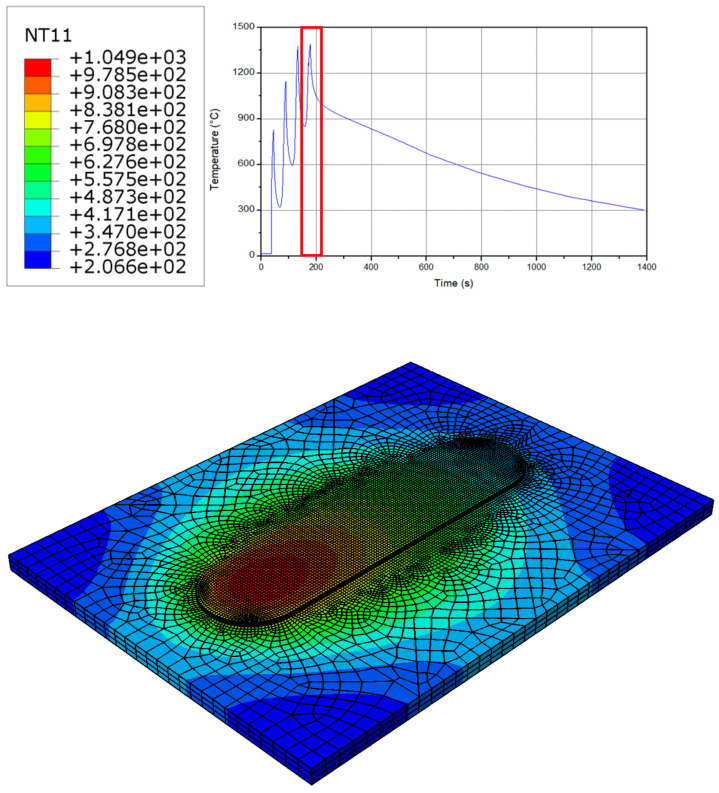
The temperature distribution at the workpiece in the fourth layer.

**Figure 6 micromachines-15-00303-f006:**
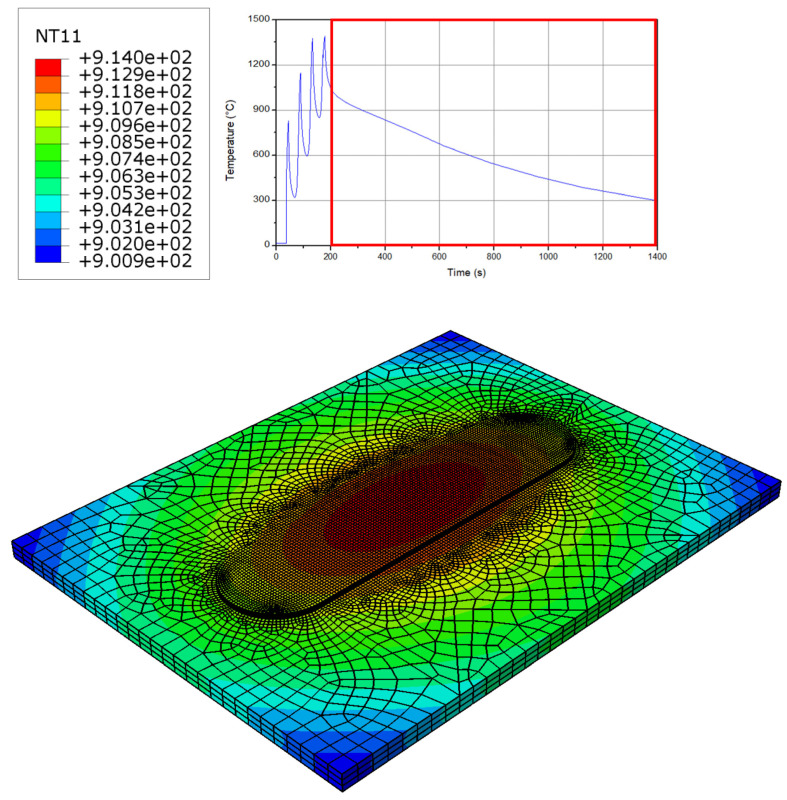
Temperature at the cooling step of the process.

**Figure 7 micromachines-15-00303-f007:**
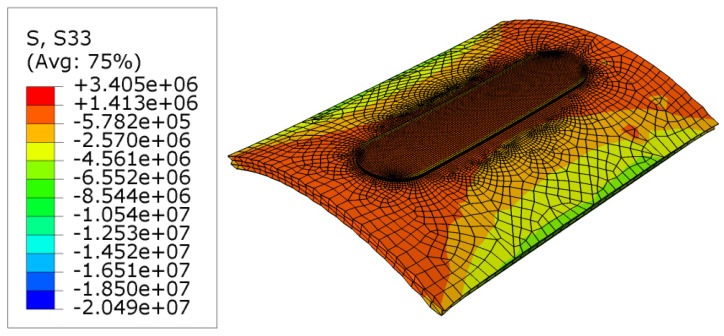
Stress distribution during the last step of the process.

**Figure 8 micromachines-15-00303-f008:**
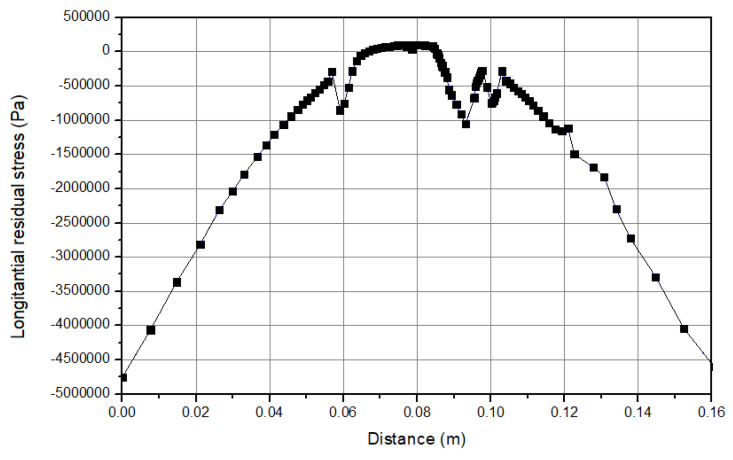
Stress distribution (longitudinal residual stress).

**Figure 9 micromachines-15-00303-f009:**
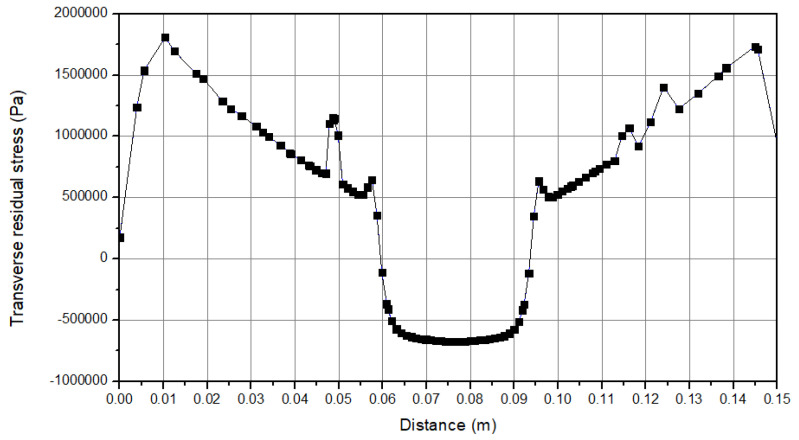
Stress distribution (transverse residual stress).

**Figure 10 micromachines-15-00303-f010:**
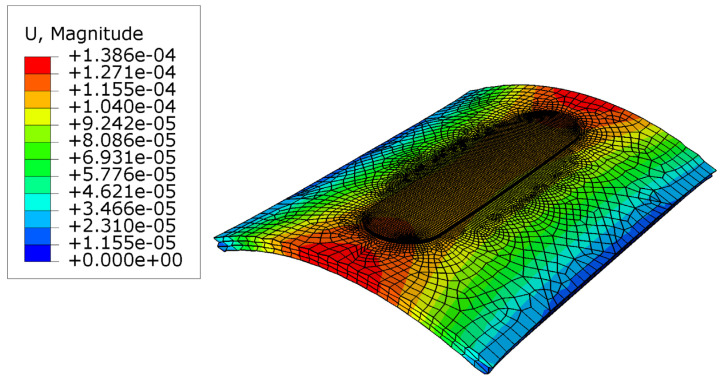
Distortion results at the last step.

**Table 1 micromachines-15-00303-t001:** Temperature-dependent material properties of AZ31.

Temperature (°C)	Elasticity Modulus (E)	Poisson’s Ratio [ν]	Thermal Expansion Coefficient
20	40.2	0.330	7.7 × 10^−6^
150	35.35	0.334	2.64 × 10^−6^
300	1.1	0.336	2.7 × 10^−6^
400	0.82	0.337	2.7 × 10^−6^
550	0.7	0.337	2.95 × 10^−6^

**Table 2 micromachines-15-00303-t002:** Temperature-dependent thermal properties of AZ31.

Temperature [T] (°C)	Thermal Conductivity [k] (W/m °C)
20	96.4
100	101
200	105
250	107
300	109
400	113
420	114
440	115
460	116

**Table 3 micromachines-15-00303-t003:** Temperature-dependent thermal properties of AZ31.

Temperature (°C)	Specific Heat [C] (J/kg °C)
20	1050
100	1130
200	1170
300	1210
350	1260
400	1300
450	1340
470	1340
500	1360

## Data Availability

Data are contained within the article.

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
