# Peer review of "A Framework to Simulate Friction Stir Additive Manufacturing (FSAM) Using the Finite Element Method"

_micromachines, 2024, doi:10.3390/mi15030303_

Round 1

Reviewer 1 Report

Comments and Suggestions for Authors

In this manuscript, a modified friction model developed in [27] (page 6) is implemented in order to simulate the sliding and sticking conditions during the FSAM process. Magnesium alloy was considered as test material. Friction model was applied to investigate the thermomechanical behaviour of the FSAM process. The results showed a significant influence of the sliding/sticking condition on the accuracy of the numerical model.

The main problem of this article is that the numerical results have not been validated experimentally. So, I don't consider the simulation results to be relevant.

The abstract does not represent the content of the article. The type of workpiece material and other important details were not even presented.

The scientific merits and main innovations of this work should be highlighted in the Abstract and Conclusions sections.

All cited items in the Introduction section require revision. For example [17] was not written by Stubblefield (line 84). I found no information in [18] that re-stirring behavior was studied.

Workpiece material has a key impact on the course of FSAM. In the Introduction section, the authors discussed various phenomena accompanying the FSAM process based on extensive literature. However, it was not presented what specific research material these particular works concern.

What do the dots in equations (1) and (2) mean?

Figure 6: Longitantial stress?

Editing of English language and style is required. Furthermore, please ensure that the appropriate grammatical tense is used consistently.

In Abstract section, the authors say "Finally the validation of the model is done by the presented results in the literature." Where in the article can I find the validation results?

The Introduction section specifies what has been done in this work. However, the aim of the work was not specified.

The Tables and Figures are appropriate and clear. It is suggested to rearrange Figures 2-6 in order to reduce whitespace.

Section 3: It is required to discuss the results and compare the interpretations with studies found in the literature.

In Conclusions section the authors indicated that "Finally, the results are verified and validated by the presented literature". Please specify paragraphs where the reader will find the validation results of distortion, stress distribution and temperature.

The discussions of the obtained results should be elaborated in Section 3.

Comments on the Quality of English Language

Editing of English language and style is required. Furthermore, please ensure that the appropriate grammatical tense is used consistently.

Author Response

Please find them in the attachment 

Reviewer 2 Report

Comments and Suggestions for Authors

1 Add a transition section to Section 2.

2 Is Figure 1 means the temperature change at a certain point over time? Suggest explanatory notes in the text and marking each layer in this figure.

3 Verify the data in the text. For example: line 199: The second layer has exceeded 1000℃, line 205: The last layer has exceeded 1350℃

4 Figures 2 to 4 should be merged into four subgraphs of one graph to compare the temperature distribution between different layers

5 Figures 8 and 9 can be represented by two subgraphs.

6 Please add some discussion content based on the results. For example, provide guidance and optimization of the process.

Author Response

Please find in the attachment

Round 2

Reviewer 1 Report

Comments and Suggestions for Authors

The authors introduced changes to the manuscript as suggested by the reviewer. I would like to thank the authors for responding to my comments. I consider them satisfying and, by the way, I congratulate you on your good publication.
I suggest accepting this manuscript in current form.